# Utilization of modern contraceptive methods and its determinants among youth in Myanmar: Analysis of Myanmar Demographic and Health Survey (2015-2016)

**Ciin Ngaih Lun** [1‡], **Thida Aung** [2☯], **Kyaw Swa Mya** [3☯]*

**1** Mingalar Taung Nyunt Township Public Health Department, Yangon, Myanmar, **2** Department of Population and Family Health, University of Public Health, Yangon, Myanmar, **3** Department of Biostatistics and Medical Demography, University of Public Health, Yangon, Myanmar

☯ These authors contributed equally to this work.
‡ This author contributed mainly to this work.
* kyawswamya@gmail.com

**Data Availability Statement:** The Myanmar Demographic and Health Survey data can be downloaded from this URL - https://dhsprogram.com/data/available-datasets.cfm.

## Abstract

Reproductive health service is crucial for youth to reduce maternal and child mortality. However, many young women face unintended pregnancies and pregnancy-related complications due to insufficient knowledge of contraceptive methods and low contraceptive utilization. This study aims to assess the modern contraceptive prevalence rates among youth and identify factors influencing modern contraceptive utilization among youth. We used Myanmar Demographic and Health Survey (2015–2016) data. This study included 1,423 men and 3,677 women aged 15–24 years from all states and regions of Myanmar. We used multivariable binary logistic regression analysis and reported the results using adjusted Odds Ratios (AOR) with 95% Confidence Intervals (CI). Data analysis was done by STATA software (version 15.1). Ever-married youth used mainly injectable contraception, followed by oral contraceptive pills. Never-married male youth mainly used oral contraceptive pills; however, almost all never-married female youth did not use contraception. The modern contraceptive prevalence rates were 14.9% among total youth, 10% among males, 16.8% among females, 1.5% among never-married males, 44.7% among ever-married males, and 54% among ever-married female youth. The knowledge on modern contraceptive methods favored the utilization. Sexually active youth utilized more contraception than sexually inactive youth. We also found geographical variation and low utilization among rural youth. The desire for more children was also a significant predictor of contraceptive utilization among married youth. The utilization of modern contraception was low among Myanmar youth. Reproductive health program needs to be emphasized on the youth population especially in the area with low utilization to have equitable access to quality reproductive health services. Moreover, the revitalization of Youth Information Corner and youth-friendly reproductive health education programs should be implemented to increase reproductive health knowledge and prevent unsafe sex, unintended pregnancies, and abortions which might help in reducing maternal and child mortality. We warranted conducting

**Funding:** The author(s) received no specific funding for this work.

**Competing interests:** The authors have declared that no competing interests exist.

mixed method studies to explore the barriers and challenges of contraceptive utilization and male involvement in the choice of contraception among youth.

## Introduction

Youth has been defined as the person between 15 and 24 years of age by the United Nations since 1981. It is the critical transitional period, from childhood dependence to adulthood independence, during which physical, psychological, social, and sexual changes occur [1]. Youth is the time when most people start exploring their sexuality and having an intimate relationship [2]. Sustainable development goals (SDGs) Goal 3 under Target 3.7 stated, "By 2030, ensure universal access to sexual and reproductive health-care services, including for family planning, information and education, and the integration of reproductive health into national strategies and programs" [3]. Therefore, youth must have the right to get quality sexual and reproductive health services and make decisions free from any violence, discrimination, and coercion [4].

Globally, pregnancy in young people has become a major public health issue for youth health. About 10% to 40% of young females encounter unwanted pregnancies in developing countries [5]. Young women who start their first sexual intercourse at young ages are less likely to use contraceptive methods due to a lack of knowledge and access to contraceptive methods [6, 7]. Getting the proper knowledge of contraception before starting the first sexual activity is essential for youth to prevent wrong decision-making, sexually transmitted infections, and unwanted pregnancy. Some studies have projected that if there were effective contraception usage among youth, it would reduce unintended pregnancies by 59%, unplanned births by 62%, abortion by 57%, miscarriage of unintended pregnancies by 71%. Moreover, nearly 32% of maternal mortality, 90% of abortion-related, and 20% of pregnancy-related morbidity and mortality could be prevented [8, 9]. Therefore, knowledge and utilization of contraception among youth play an essential role in saving women's lives by reducing unwanted and high-risk pregnancies, unsafe abortion, and maternal mortality and improving the survival rates of newborn children by lengthening the interval between pregnancies [4].

In Myanmar, having sexual activity among youth aged 15–19 years was about 12% among girls and 7% among boys. In youth aged 20–24 years, the practice was about 48% among girls and 43% among boys [10]. The adolescent birth rate is 33 per 1,000 women aged 15–19 years, and the age-specific fertility rate for 20–24 years old is 108 per 1,000 women [11]. The ratios of newborn infant death to age group-specific births were about 6% in 15–19 years age group mothers and 4.6% in the age groups of 20–24 years old mothers [12].

The Ministry of Health and Sports (MOHS), Myanmar, in collaboration with UNFPA and WHO, published the Five-Year Strategic Plan for Young People's Health (2016–2020). Four targets related to reproductive health were: 1) to reduce adolescent fertility rate from 20 per 1,000 in 2014 to 10 per 1,000 in 2018; 2) to increase the contraceptive prevalence rate among sexually active young people from 38% in 2014 to 52% in 2018; 3) to reduce MMR(Maternal Mortality Rate) among young pregnant mother and 4) to increase the proportion of young people with correct knowledge of Sexual and Reproductive Health and HIV/AIDS [13]. To meet these mandates, Myanmar policymakers and planners need a broad understanding of sexual and reproductive health issues, such as utilizing modern contraceptives among Myanmar youth, their determinants, and how they vary according to sex.

In Myanmar, the Youth Information Corners (YICs) have been established since 2002 by the Health Education Division under the Ministry of Health, supporting UNFPA to promote

reproductive health knowledge among youth [14]. YICs began with 17 rural health centers in 2002 and eventually reaching 70 health centers across the country in 2012. YICs act as a library providing literature on reproductive health and other youth-related information, edutainment devices for youth. YICs also aimed to train youth volunteers to become peer educators on reproductive health issues. However, most of the YICs had stopped operating at the time of situation analysis. Hence, the Five-Year Strategic Plan for Young People's Health recommended an immediate rejuvenation of the YIC program [13].

There are some Myanmar studies regarding contraceptive utilization [15–19]. These studies were conducted among men, reproductive-aged women, and married migrant women and not focused on the youth population. Mon et al. conducted a study among married youth and their husband to assess the contraceptive utilization [20], and Lat et al. also conducted a study among never-married youth regarding premarital sex [21]; however, these studies used a small sample to represent a particular township rather than the whole country. Therefore, we conducted this study to assess the utilization of modern contraception and its determinants among Myanmar youth.

## Materials and methods

We used the Myanmar Demographic and Health Survey (MDHS) data, a nationally representative population-based survey conducted during 2015–16. The detailed methodology of this survey was published elsewhere [10]. It was a nationwide cross-sectional study that allowed estimates of key indicators at the national level, urban and rural areas, and all states and regions of Myanmar. The sample was a stratified two-stage cluster sample of households that included 442 clusters (123 from urban areas and 319 from rural areas): 30 households from each cluster to get 13,260 households using equal probability systematic sampling. All women aged 15–49 years from selected households and all men aged 15–49 years from a subsample consisting of one household in every second household selected for the female survey were eligible. A total of 12,885 women and 4,737 men were interviewed in the survey. From those, we included 1,423 men and 3,677 women age 15–24 years in this study.

### Dependent variable

We used the utilization of modern contraceptive methods among Myanmar youth as a dependent variable. The modern contraceptive methods used in MDHS's annual report are as follow– 1) male sterilization, 2) female sterilization, 3) intrauterine device (IUD), 4) contraceptive implant, 5) injectable contraception, 6) oral contraceptive pill, 7) male condom, 8) female condom, 9) emergency contraception, 10) lactational amenorrhea method and 11) others (cervical cap and sponge) [10]. If the respondent or sexual partner currently used any type of modern contraception at the time of the survey, he or she was categorized as used modern contraceptives coded as "yes" and, if not, categorized as did not use modern contraceptives coded as "no".

### Independent variables

We used the number of modern contraceptive methods known by youth as an independent variable. We gave score-1 for every method known by youth. Hence, the possible score ranged from 0 to 11, and we treated this variable as a continuous variable. Moreover, we also used the individual and household characteristics of youth as independent variables. The individual characteristics were age, sex, marital status, education, employment, previous exposure to family planning messages, sexual activity, and desire for more children. We categorized the age into 15–19 years and 20–24 years and sex into male and female. The marital status was

categorized into never-married and ever-married, which included currently married, widowed, separated. The education level was categorized into no education, primary, secondary and above.

The employment variable was recoded as "yes" if the respondent had an earning job within 12 months before the survey or "no" otherwise. The exposure to family planning messages variable was recoded as "yes" if he or she heard about family planning messages from any one of TV, radio, newspaper, internet, billboard, health care provider within 12 months before the survey. If they had not heard any family planning messages, they were recoded as "no". Sexually activity was recoded as "active" if the respondent had sexual activity within four weeks before the survey, and if not, it was recoded as "not active". The information of desire for more children was available only for ever-married youth. It was categorized into "yes" if the youth desired to get more children and "no" if they did not want any more children or fecund or never had sex.

The household characteristics were the residence (categorized into urban and rural residence), geographical zone (categorized into the hilly, coastal, delta, and central plain), and wealth index (categorized into rich, middle, and poor).

## Statistical analysis

Data were analyzed using STATA software (version 15.1). We used the survey data analysis command (svy) to adjust the cluster survey design and non-response rates for each analysis. All estimates were weighted to represent the whole population in the nation. We described the modern contraceptives prevalence rate (mCPR) with 95% error bars for total youth, ever-married youth, never-married youth, male youth, and female youth. Frequency distribution tables reported the background characteristics of the sample. We used the Chi-square test of independence to assess the relationship between independent variables and dependent variable. We checked the multicollinearity; however, we did not find any highly correlated independent variables.

We used the multivariable binary logistic regression analysis to assess the factors influencing the utilization of modern contraceptive methods. We analyzed the data separately for never-married and ever-married youth for each sex to distinguish the marital status's effect on contraceptive utilization. Since the utilization was extremely low among never-married females (only one female youth utilized contraceptives), it was impossible to analyze for this group. Hence, we did not report for never-married females in the Results section. Moreover, we assessed the gender's effect on utilization by analyzing separately for male youth and female youth without considering the marital status and the marital status's effect on utilization by analyzing separately for never-married youth and ever-married youth without considering the gender difference. We reported the results using adjusted odds ratios (AOR) with 95% confidence intervals. P-value <0.05 was set as a statistical significance.

**Ethical consideration.** The MDHS protocol was approved by the Ethics Review Committee on Medical Research, including Human Subjects in the Department of Medical Research, the Ministry of Health and Sports, and the ICF Institutional Review Board, respectively. A verbal informed consent was taken from each participant before interview. MDHS data are fully anonymized so that all individual identifiers could not be accessible by data users. We obtained the Ethical approval for this study from the Institutional Review Board of the University of Public Health (UPH-IRB (2020/MPH/2)).

## Results

We reported the background characteristics of youth in Table 1. The mean age of married youth was older than that of unmarried youth for both sexes (22 vs. 19 years for males and

**Table 1. The individual and household characteristics of youth in Myanmar.**

| Variables | Male youth (N = 1,423) | | Female youth (N = 3,677) | |
|---|---|---|---|---|
| | Never-married (n = 1,143) | Ever-married (n = 280) | Never-married (n = 2,533) | Ever-married (n = 1,144) |
| Individual characteristics | | | | |
| Age | | | | |
| 15–19 years | 693 (60.6) | 39 (13.9) | 1564 (61.8) | 246 (21.5) |
| 20–24 years | 450 (39.4) | 241 (86.1) | 969 (38.2) | 898 (78.5) |
| Mean ± SD | 19 ± 2.7 | 22 ± 1.9 | 19 ± 2.8 | 21 ± 2.2 |
| Education | | | | |
| No education | 81 (7.1) | 28 (9.9) | 122 (4.8) | 142 (12.4) |
| Primary | 235 (20.5) | 103 (36.7) | 575 (22.7) | 438 (38.3) |
| Secondary and higher | 827 (72.4) | 149 (53.4) | 1,836 (72.5) | 564 (49.3) |
| Employment | | | | |
| No | 290 (25.4) | 7 (2.4) | 989 (39.1) | 515 (45.0) |
| Yes | 853 (74.6) | 273 (97.7) | 1,543 (60.9) | 629 (55.0) |
| Exposure to family planning messages | | | | |
| No | 547 (47.9) | 132 (47.2) | 1,368 (54.0) | 668 (58.4) |
| Yes | 596 (52.1) | 148 (52.8) | 1,165 (46.0) | 476 (41.6) |
| Sexually active | | | | |
| No | 1,123 (98.2) | 40 (14.4) | 2,533 (100.0) | 282 (24.7) |
| Yes | 20 (1.8) | 240 (85.6) | 0 (0.0) | 862 (75.3) |
| Desire for more children [†] | | | | |
| Yes | | 202 (72.1) | | 701 (61.3) |
| No | | 78 (27.9) | | 443 (38.7) |
| Household characteristics | | | | |
| Residence | | | | |
| Urban | 369 (32.3) | 73 (26.0) | 845 (33.4) | 276 (24.1) |
| Rural | 774 (67.7) | 207 (74.0) | 1,688 (66.6) | 868 (75.9) |
| Geographical zone | | | | |
| Hilly | 185 (16.2) | 70 (24.8) | 450 (17.8) | 247 (21.6) |
| Coastal | 155 (13.6) | 28 (10.0) | 324 (12.8) | 168 (14.7) |
| Delta | 421 (36.8) | 104 (37.4) | 967 (38.2) | 412 (36.0) |
| Central plain | 382 (33.4) | 78 (27.8) | 792 (31.2) | 317 (27.7) |
| Wealth index | | | | |
| Poor | 392 (34.3) | 114 (40.9) | 788 (31.1) | 531 (46.4) |
| Middle | 255 (22.4) | 65 (23.2) | 553 (21.9) | 224 (19.5) |
| Rich | 496 (43.4) | 101 (35.9) | 1192 (47.1) | 389 (34.1) |

The results were presented by n (%). SD: Standard Deviation,

[†]This information was only available for ever-married youth in MDHS.

The data of responding "not sure, never had sex and missing values" were added to the "No desire for more children" category group.

21 vs. 19 years for females). The number of youth who passed secondary and higher education among unmarried was more than that of married youth for both sexes (72.4% vs. 53.4% for male and 72.5% vs. 49.3% for female). Almost all married male youth were employed; however, about half of married female youth were employed. Male youth had higher exposure to family planning messages than female youth for both unmarried and married youth.

None of the never-married female youth was sexually active, and only 1.8% of never-married male youth were sexually active. In comparison, 85% of males and 75% of females among

married youth were sexually active. Most of the youth were from the rural area, i.e., about two-third of never-married youth and three-fourths of ever-married youth. Delta zone had the highest number of youth, and the coastal zone had the lowest number of youth for both sexes and all marital statuses. Almost half of the never-married youth were from rich households (43% for male and 47% for female); however, nearly half of the ever-married youth were from poor households (41% for male and 46% for female).

We described the modern contraceptive methods known by youth in Fig 1. Among never-married male youth, the male condom was the most known method (82%), followed by oral contraceptive pill (72%) and injectable contraception (71%). Among never-married female youth, injectable contraception (88%) was the most known method, followed by oral contraceptive pills (86%) and female sterilization (73%). See detail in Fig 1a.

Among ever-married male youth, injectable contraception was the most known method (89%), followed by oral contraceptive pills (88%) and male condoms (83%). Among ever-married female youth, injectable contraception (96%) was the most known method, followed by oral contraceptive pills (94%) and female sterilization (80%). See detail in Fig 1b.

We reported the total numbers of modern contraceptive methods known by youth in Table 2. Among never-married youth, most male youth (17%) knew four methods, while most female youth (17%) knew six modern contraception methods. Among ever-married youth, male youth (19%) mostly knew five methods while female youth (16%) mostly knew six modern contraception methods. Youth who did not know any contraceptive method were 10% among never-married male youth, 7% among never-married female youth, 6% among ever-married male youth, and 3% among ever-married female youth.

We described the types of modern contraceptive methods used by youth in Table 3. Among never-married male youth, the oral contraceptive pill was mostly used (1.1%), followed by injectable contraception (0.2%) and others (cervical cap and sponge) (0.2%). Among ever-married male youth, injectable contraception was mostly used (25%), followed by oral contraceptive pill (18%), intrauterine device (0.6%), and female sterilization (0.6%). Almost all of the never-married female youth (99.97%) did not use any method of contraception, and only 0.03% used oral contraceptive pills. Among ever-married female youth, injectable contraception was mostly used (33%), followed by oral contraceptive pill (19%), intrauterine devices (0.9%), contraceptive implants (0.6%), female sterilization (0.3%), male condom (0.2%) and others (0.08%).

Fig 2 shows the mCPR of Myanmar youth. The mCPR was 14.9% [95%CI: 13.6, 16.3] among all youth, 10% [95%CI: 8.4, 11.9] among male youth, and 16.8% [95%CI: 15.3, 18.5] among female youth. It was 52.2% [95%CI: 48.9, 55.4] among all ever-married youth, 44.7% [95%CI: 37.6, 52.0] among ever-married male youth and 54% [95%CI: 50.6, 57.4] among ever-married female youth. It was 0.5% [95%CI: 0.3, 0.9] among all never-married youth, 1.5% [95%CI: 0.8, 2.7] among never-married male youth; however, only one never-married female youth utilized modern contraception.

We performed the test for trend across the ordered group to assess whether there is a trend in mCPR as increased in the number of modern methods known among total youth (n = 5100). We found a significant increasing trend (p<0.001), i.e., as the number of methods known increased, the mCPR also increased. See detailed in Fig 3.

We assessed the relationship between modern contraceptive methods utilization and background characteristics and reported in Table 4. Since almost all never-married female youth did not utilize modern contraception, we did not report for never-married female youth. Older youth (20-24years) had higher modern contraceptive utilization than younger youth (15-19years); however, it was significant only for never-married male youth (3.2% vs. 0.4%). Regarding educational status, we found the increased utilization trend as the

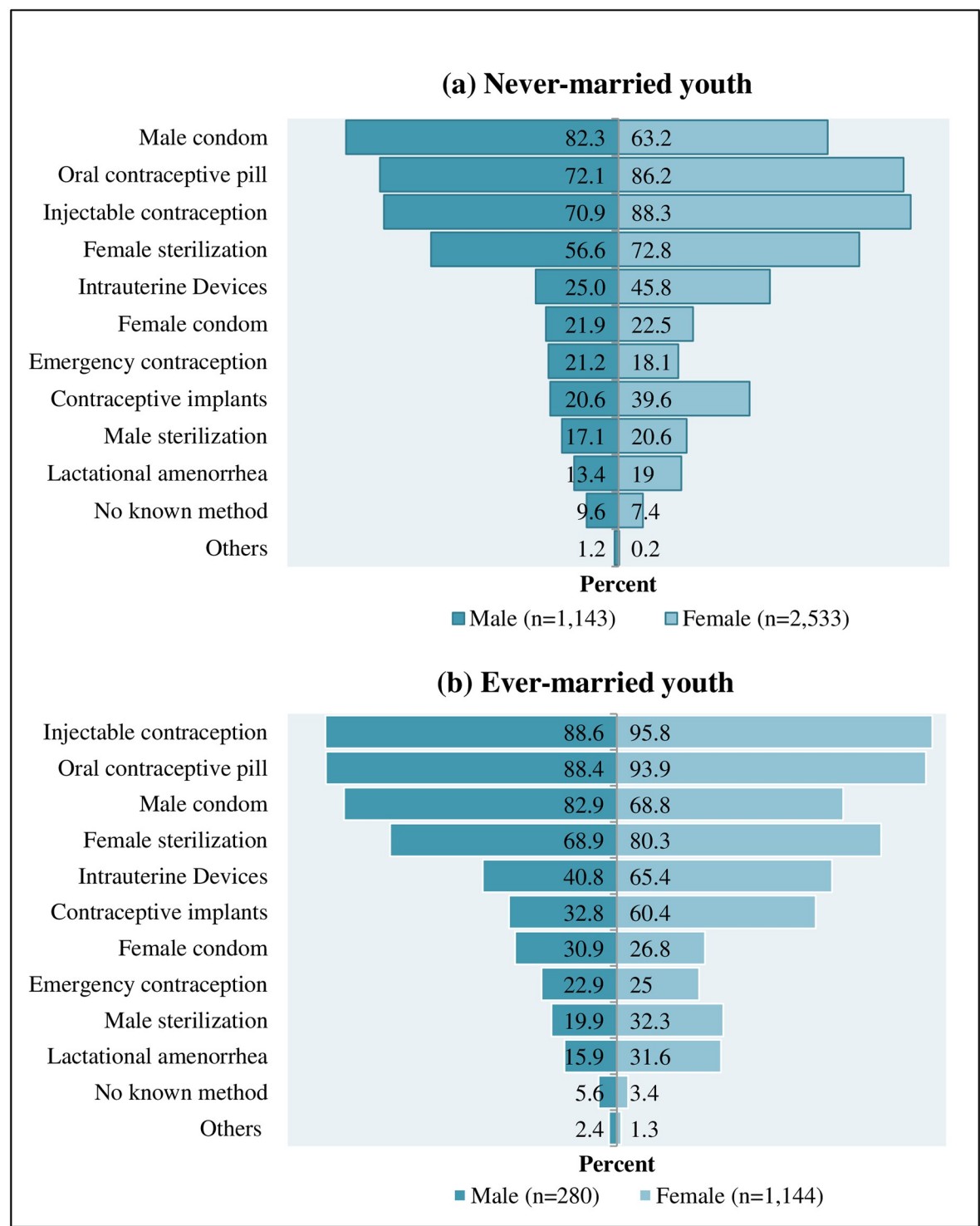

**Fig 1. Types of modern contraceptive methods known by youth in Myanmar.**

education status increased only among ever-married female youth (40.4% in no education, 51.1% in primary education, and 54.9% in secondary and higher education). Sexually active youth utilized contraception significantly more than sexually not active youth in all groups.

**Table 2. Number of modern contraceptive methods known by youth in Myanmar.**

| Number of modern contraceptive methods known by the youth | Male youth (N = 1,423) | | Female youth (N = 3,677) | |
|---|---|---|---|---|
| | Never-married (n = 1,143) | Ever-married (n = 280) | Never-married (n = 2,533) | Ever-married (n = 1,144) |
| 0 | 110 (9.6) | 16 (5.6) | 188 (7.4) | 39 (3.4) |
| 1 | 97 (8.5) | 11 (4.0) | 85 (3.4) | 20 (1.8) |
| 2 | 107 (9.4) | 17 (6.3) | 178 (7.0) | 86 (7.5) |
| 3 | 166 (14.5) | 26 (9.2) | 277 (11.0) | 80 (7.0) |
| 4 | 194 (17.0) | 44 (15.8) | 373 (14.7) | 107 (9.4) |
| 5 | 164 (14.3) | 53 (18.9) | 407 (16.1) | 130 (11.3) |
| 6 | 111 (9.7) | 42 (15.0) | 422 (16.7) | 183 (16.0) |
| 7 | 86 (7.5) | 28 (9.9) | 288 (11.4) | 182 (15.9) |
| 8 | 62 (5.4) | 23 (8.1) | 177 (7.0) | 153 (13.4) |
| 9 | 35 (3.0) | 13 (4.8) | 86 (3.4) | 112 (9.8) |
| 10 | 11 (1.0) | 7 (2.5) | 52 (2.0) | 48 (4.2) |
| 11 | 0 (0) | 0 (0) | 0 (0) | 4 (0.3) |

The results were presented by n (%). Six missing values were added to no known contraceptive method group.

Youth from urban households utilized more contraception than rural households; however, we did not find this urban-rural difference among ever-married male youth. We found a significant regional variation in utilization of modern conception among ever-married male and female youth. The utilization was highest among youth from the delta zone and lowest among youth from the hilly zone. The household wealth status also influenced the modern contraceptive utilization among never-married youth only. We found the highest utilization among youth from the rich households and lowest among youth from poor households (2.5% vs. 0.3%).

We used multivariable binary logistic regression to assess the determinants of modern contraceptive utilization among never-married male youth, ever-married male youth, and ever-married female youth, adjusting the covariates (Table 5). Among never-married male youth, sexual activity and the number of known modern contraceptive methods were significant predictors of utilizing modern contraception. Sexually active youth utilized modern contraception 78 times more than those who were not sexually active. As youth who knew one more modern contraception, the odds of modern contraceptive utilization was 38% increased.

**Table 3. Types of modern contraceptive methods used by youth in Myanmar.**

| Contraceptive methods used by Myanmar youth | Male (N = 1,423) | | Female (N = 3,677) | |
|---|---|---|---|---|
| | Never-married (n = 1,143) | Ever-married (n = 280) | Never-married (n = 2,533) | Ever-married (n = 1,144) |
| No method | 1,121 (98.0) | 155 (55.3) | 2,532 (99.97) | 523 (45.7) |
| Injectable contraception | 2 (0.2) | 70 (25.2) | 0 (0.0) | 378 (33.1) |
| Oral contraceptive pill | 13 (1.1) | 51 (18.2) | 1 (0.03) | 216 (18.9) |
| Intrauterine devices | 0 (0.0) | 2 (0.6) | 0 (0.0) | 10 (0.9) |
| Contraceptive implants | 0 (0.0) | 0 (0.0) | 0 (0.0) | 6 (0.6) |
| Female sterilization | 0 (0.0) | 2 (0.6) | 0 (0.0) | 4 (0.3) |
| Male condom | 0 (0.0) | 0 (0.0) | 0 (0.0) | 2 (0.2) |
| Others | 2 (0.2) | 0 (0.0) | 0 (0.0) | 1 (0.08) |

Multiple responses, Presented by n (%).

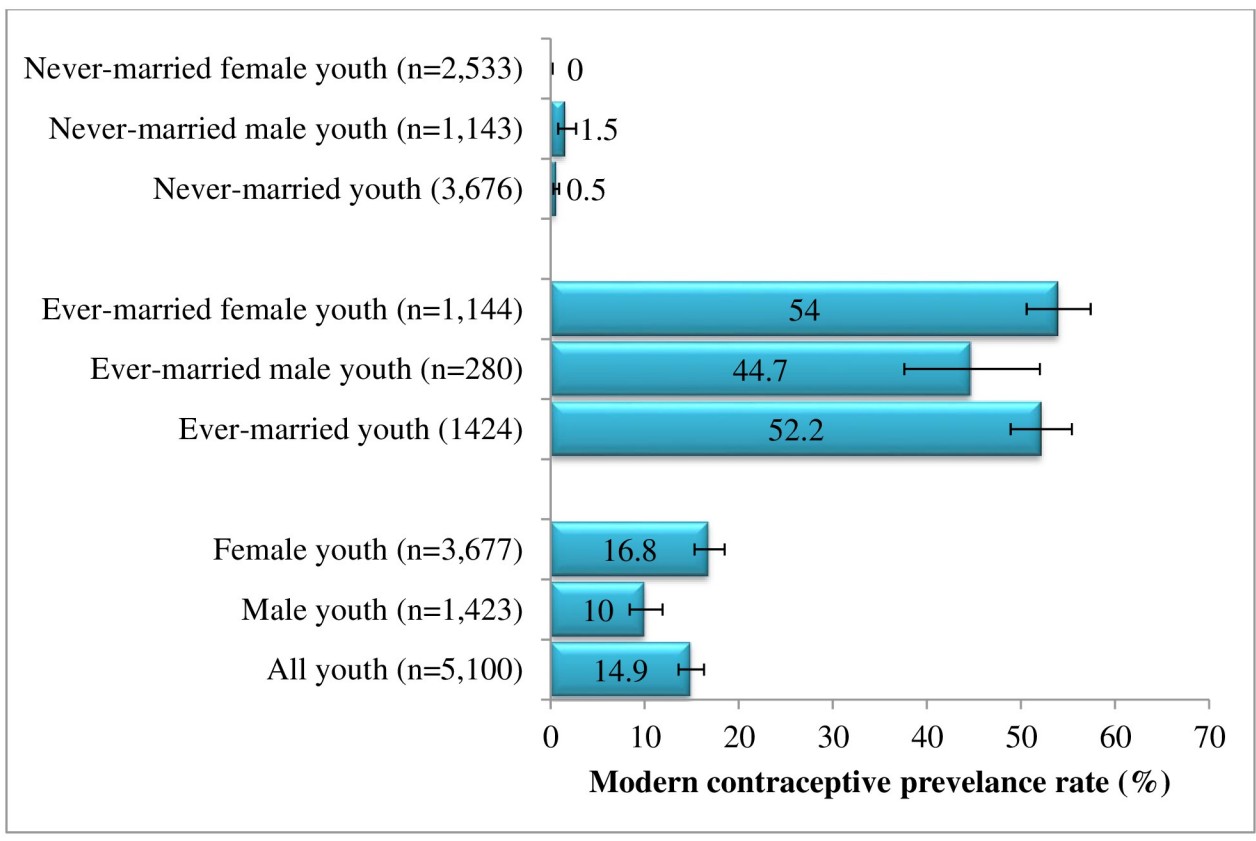

**Fig 2. Prevalence of modern contraceptive utilization among youth in Myanmar.**

Among ever-married male youth, sexual activity and geographical zone were significant predictors of modern contraceptive utilization. Sexually active youth were 3.5 times more likely to use modern contraception than those who were not sexually active. Youth from the hilly zone had 70% fewer odds of using modern contraception than those from the delta zone.

Among ever-married female youth, sexual activity, desire for more children, number of known modern contraceptive methods, and geographical zone were significant predictors of modern contraceptive utilization. Sexually active youth were seven times more likely to use modern contraception than sexually not active youth. Youth who did not want more children were 49% more likely to use modern contraception than youth who want more children. Increased in knowing one more modern contraceptive method had 18% higher odds of modern contraceptive utilization among ever-married female youth. Youth living in rural areas were 31% less likely to utilize modern contraception than urban areas. Regional differences of ever-married female youth were significantly associated with differences in modern contraceptive utilization. Youth from the hilly zone, coastal zone, and central plain zone were less likely to use modern contraception than youth from the delta zone by 71%, 63%, and 44%.

We reported the multivariable binary logistic regression analysis of modern contraceptive methods utilization among male, female, ever-married, and never-married youth in Table 6, adjusting the covariates. Among total male youth, age, sexual activity, number of known modern contraceptive methods, and geographical zone were significant predictors of modern contraception. Older male youth had two times more likely to use modern contraception than

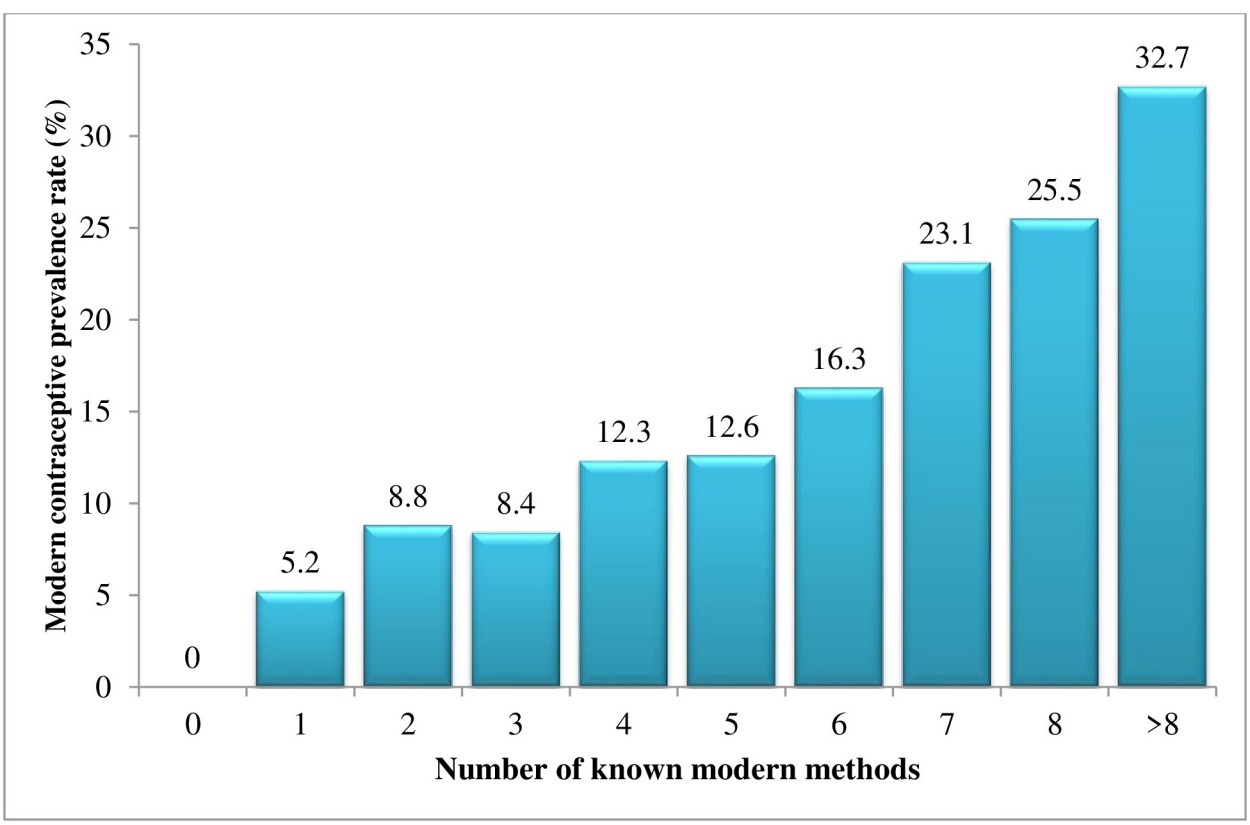

**Fig 3. The modern contraceptive prevalence rate by the number of known methods among youth in Myanmar.**

younger male youth. The utilization of sexually active male youth was 41 times more than sexually not active male youth. Increased knowledge of one more modern contraception method had an 18% higher chance of modern contraceptive utilization among male youth. Youth from the hilly zone had less odds of modern contraceptive utilization than youth from the delta zone by 59%.

Among total female youth, age, sexual activity, number of known modern contraceptive methods, residence, geographical zone, and wealth status were significant predictors of modern contraception. Older female youth were 1.79 times more likely to use modern contraception than younger female youth. Female youth who were sexually active utilized 54 times more than those who were not sexually active. Increased in knowing one more method increased 23% higher odds of modern contraceptive utilization. Female youth from rural areas were 35% less likely to use modern contraception than urban areas. Youth from the hilly zone, coastal zone, and central plain zone had less odds of modern contraceptive utilization than youth from the delta zone by 63%, 58%, and 36%. Female youth from rich households were 32% less likely to use modern contraception than poor households.

Among ever-married youth, sexual activity, desire for more children, the number of known modern contraceptive methods, residence, and geographical zone were significant predictors of modern contraceptive utilization. Sexually active youth were six times more likely to use modern contraceptives than those who were not sexually active. Youth who did not want more children used modern contraception 1.5 times more than those who want more children. One more known modern contraceptive method increased modern contraceptive utilization by

**Table 4. The association between modern contraceptive methods utilization by background characteristics among youth in Myanmar (Bivariate analysis).**

| Variables | Male youth(n = 1,423) | | Female youth(n = 3,677) |
|---|---|---|---|
| | Never-married (n = 1,143) | Ever-married (n = 280) | Ever-married (n = 1,144) |
| | mCPR% [95%CI] | mCPR% [95%CI] | mCPR%[95%CI] |
| Individual characteristics | | | |
| Age | p = 0.001 | p = 0.135 | p = 0.142 |
| 15–19 years | 0.4 [0.1–1.2] | 32.0 [17.7, 50.8] | 49.2 [42.0, 56.4] |
| 20–24 years | 3.2 [1.7, 6.0] | 46.7 [39.2, 54.4] | 55.3 [51.4, 59.2] |
| Education | p = 0.541 | p = 0.090 | p = 0.003 |
| No education | 1.0 [0.1, 7.2] | 21.7 [7.0, 50.6] | 40.4 [30.2, 51.5] |
| Primary | 0.7 [0.1, 4.4] | 41.8 [30.2, 54.4] | 51.1 [45.2, 56.8] |
| Secondary and higher | 1.8 [0.9, 3.3] | 51.0 [41.8, 60.1] | 59.9 [54.9, 64.7] |
| Employment | p = 0.798 | p = 0.311 | p = 0.774 |
| No | 1.3 [0.5, 3.8] | 64.2 [26.7, 89.8] | 53.4 [48.4, 58.4] |
| Yes | 1.6 [0.8, 3.1] | 44.2 [37.0, 51.7] | 54.5 [49.6, 59.3] |
| Exposure to family planning messages | p = 0.357 | p = 0.128 | p = 0.097 |
| No | 1.1 [0.4, 3.0] | 38.5 [28.8, 49.2] | 51.3 [46.8, 55.8] |
| Yes | 1.9 [0.9, 3.8] | 50.3 [39.8, 60.7] | 57.8 [52.0, 63.4] |
| Sexually active | p<0.001 | p = 0.001 | p<0.001 |
| No | 0.7 [0.3, 1.5] | 15.9 [7.6, 30.2] | 21.7 [16.6, 27.7] |
| Yes | 46.2 [21.3, 73.2] | 49.5 [41.7, 57.4] | 64.6 [60.6, 68.4] |
| Desire for more children[†] | | p = 0.393 | p = 0.205 |
| Yes | | 46.8 [38.8, 55.0] | 52.2 [47.4, 57.0] |
| No | | 39.3 [25.7, 54.7] | 56.9 [51.7, 61.9] |
| Household characteristics | | | |
| Residence | p = 0.017 | p = 0.960 | p = 0.014 |
| Urban | 3.0 [1.4, 6.3] | 44.4 [30.5, 59.2] | 61.3 [54.8, 67.4] |
| Rural | 0.8 [0.3, 1.9] | 44.8 [36.7, 53.2] | 51.7 [47.7, 55.7] |
| Geographical zone | p = 0.603 | p = 0.004 | p<0.001 |
| Hilly | 1.5 [0.3, 6.9] | 25.1 [13.7, 41.4] | 34.5 [28.1, 41.4] |
| Coastal | 0.8 [0.2, 3.5] | 48.1 [33.1, 63.4] | 39.5 [32.5, 46.9] |
| Delta | 2.2 [0.9, 5.0] | 58.1 [46.1, 69.3] | 71.9 [66.0, 77.2] |
| Central plain | 1.1 [0.4, 2.8] | 42.9 [30.7, 56.2] | 53.7 [46.8, 60.4] |
| Wealth index | p = 0.042 | p = 0.085 | p = 0.923 |
| Poor | 0.3 [0.0, 1.3] | 35.8 [25.2, 47.8] | 53.3 [48.3, 58.3] |
| Middle | 1.3 [0.4, 4.5] | 58.0 [41.5, 72.8] | 55.0 [47.2, 62.5] |
| Rich | 2.5 [1.2, 5.0] | 46.3 [35.1, 57.9] | 54.4 [48.5, 60.2] |

The data of responding "not sure, never had sex and missing values" were added to the "No desire for more children" category group.

[†]This information was only available for ever-married youth in MDHS.

P values were calculated using the Chi-square test of independence.

18%. Youth living in rural areas were 23% fewer odds to use modern contraception than urban areas. Youth from the hilly zone, coastal zone, and central plain zone had less odds of modern contraceptive utilization than youth from the delta zone by 70%, 57%, and 39%.

Among never-married youth, sexual activity was the only predictor of modern contraceptive utilization. Sexually active youth were 265 times more likely to use modern contraception than those who were not sexually active.

**Table 5. Multivariable binary logistic regression analysis of modern contraceptive methods utilization among never-married male, ever-married male and ever-married female youth in Myanmar.**

| Variables | Never-married male (n = 1,143) | Ever-married male (n = 280) | Ever-married female (n = 1,144) |
|---|---|---|---|
|  | AOR [95%CI] | AOR [95%CI] | AOR [95%CI] |
| Individual characteristics |  |  |  |
| Age |  |  |  |
| 15–19 years |  |  |  |
| 20–24 years | 1.00 | 1.00 | 1.00 |
| Education | 1.95 [0.56, 6.72] | 1.49 [0.64, 3.46] | 1.12 [0.81, 1.54] |
| No education | 1.00 | 1.00 | 1.00 |
| Primary | 0.23 [0.02, 2.96] | 1.49 [0.41, 5.37] | 1.04 [0.65, 1.67] |
| Secondary and higher | 0.13 [0.01, 1.51] | 1.50 [0.40, 5.63] | 1.27 [0.76, 2.11] |
| Employment |  |  |  |
| No Yes | 1.00 | 1.00 | 1.00 |
|  | 0.62 [0.17, 2.30] | 0.80 [0.17, 3.67] | 1.00 [0.77, 1.31] |
| Exposure to family planning messages |  |  |  |
| No | 1.00 | 1.00 | 1.00 |
| Yes | 1.76 [0.39, 8.00] | 1.10 [0.60, 2.01] | 0.79 [0.58, 1.07] |
| Sexually active |  |  |  |
| No | 1.00 | 1.00 | 1.00 |
| Yes | 78.12***[17.38, 351.07] | 3.47** [1.57, 7.69] | 6.96*** [5.01, 9.69] |
| Desire for more children† |  |  |  |
| Yes |  | 1.00 | 1.00 |
| No |  | 1.16 [0.61, 2.19] | 1.49** [1.12, 2.00] |
| Number of known modern contraceptive methods | 1.38* [1.04, 1.83] | 1.12 [0.98, 1.29] | 1.18*** [1.10, 1.25] |
| Household characteristics |  |  |  |
| Residence |  |  |  |
| Urban | 1.00 | 1.00 | 1.00 |
| Rural | 0.39 [0.10, 1.53] | 0.88 [0.44, 1.72] | 0.69* [0.48, 0.98] |
| Geographical zone |  |  |  |
| Delta | 1.00 | 1.00 | 1.00 |
| Hilly | 1.52 [0.23, 9.84] | 0.30** [0.14, 0.63] | 0.29*** [0.20, 0.43] |
| Coastal | 1.44 [0.19, 10.77] | 0.68 [0.29, 1.58] | 0.37*** [0.24, 0.56] |
| Central plain | 2.75 [0.51, 14.89] | 0.75 [0.37, 1.55] | 0.56** [0.38, 0.83] |
| Wealth index |  |  |  |
| Poor | 1.00 | 1.00 | 1.00 |
| Middle | 0.98 [0.16, 5.80] | 1.47 [0.71, 3.04] | 0.79 [0.55, 1.13] |
| Rich | 0.67 [0.11, 3.99] | 0.98 [0.46, 2.11] | 0.74 [0.51, 1.08] |

***p<0.001,

**p<0.01,

*p<0.05,

AOR: adjusted odds ratio, CI: Confidence Interval.

†This information was only available for ever-married youth in MDHS.

The results of never-married female youth were not described due to very little contraceptive utilization among them.

## Discussion

The mCPR was low among Myanmar youth: 15% among total youth, 10% among males, and 16.8% among female youth. Nearly similar findings were found in Mali (15.3% in 2012) [22],

**Table 6. Multivariable binary logistic regression analysis of modern contraceptive methods utilization among male youth, female youth, ever-married youth, and never-married youth in Myanmar.**

| Variables | Male youth (n = 1,423) | Female youth (n = 3,677) | Ever-married youth (n = 1,424) | Never-married youth (n = 3,676) |
|---|---|---|---|---|
| | AOR [95%CI] | AOR [95%CI] | AOR [95%CI] | AOR [95%CI] |
| **Individual characteristics** | | | | |
| Age | | | | |
| 15–19 years | 1.00 | 1.00 | 1.00 | 1.00 |
| 20–24 years | 2.21* [1.16, 4.22] | 1.79*** [1.34, 2.39] | 1.12 [0.84, 1.51] | 2.33 [0.70, 7.74] |
| Education | | | | |
| No education | 1.00 | 1.00 | 1.00 | 1.00 |
| Primary | 1.04 [0.32, 3.32] | 0.95 [0.58, 1.56] | 1.06 [0.69, 1.65] | 0.33 [0.03, 3.63] |
| Secondary and higher | 0.90 [0.27, 2.97] | 0.82 [0.48, 1.37] | 1.21 [0.76, 1.93] | 0.16 [0.01, 1.74] |
| Employment | | | | |
| No | 1.00 | 1.00 | 1.00 | 1.00 |
| Yes | 1.10 [0.45, 2.67] | 0.84 [0.65, 1.09] | 0.90 [0.71, 1.15] | 0.61 [0.18, 2.04] |
| Exposure to family planning messages | | | | |
| No | 1.00 | 1.00 | 1.00 | 1.00 |
| Yes | 1.06 [0.62, 1.84] | 0.79 [0.59, 1.05] | 0.84 [0.64, 1.09] | 1.78 [0.45, 7.02] |
| Sexually active | | | | |
| No | 1.00 | 1.00 | 1.00 | 1.00 |
| Yes | 41.06*** [22.87, 73.71] | 54.07*** [39.89, 73.30] | 6.14*** [4.53, 8.31] | 265.34*** [61.69, 1141.37] |
| Desire for more children † | | | | |
| Yes | | | 1.00 | |
| No | | | 1.45** [1.11, 1.88] | |
| Number of known modern contraceptive methods | 1.18** [1.05, 1.34] | 1.23*** [1.16, 1.31] | 1.18*** [1.11, 1.25] | 1.23 [0.94, 1.62] |
| **Household characteristics** | | | | |
| Residence | | | | |
| Urban | 1.00 | 1.00 | 1.00 | 1.00 |
| Rural | 0.72 [0.40, 1.30] | 0.65** [0.47, 0.91] | 0.73* [0.53, 0.99] | 0.39 [0.10, 1.46] |
| Geographical zone | | | | |
| Delta | 1.00 | 1.00 | 1.00 | 1.00 |
| Hilly | 0.41** [0.21, 0.81] | 0.37*** [0.26, 0.53] | 0.30*** [0.21, 0.42] | 1.25 [0.21, 7.33] |
| Coastal | 0.86 [0.42, 1.78] | 0.42*** [0.28, 0.62] | 0.43*** [0.29, 0.62] | 2.36 [0.38, 14.55] |
| Central plain | 1.02 [0.54, 1.93] | 0.64* [0.45, 0.92] | 0.61** [0.43, 0.86] | 2.54 [0.51, 12.64] |
| Wealth index | | | | |
| Poor | 1.00 | 1.00 | 1.00 | 1.00 |
| Middle | 1.45 [0.76, 2.78] | 0.78 [0.54, 1.11] | 0.92 [0.67, 1.27] | 0.78 [0.14, 4.29] |
| Rich | 0.95 [0.48, 1.89] | 0.68* [0.47, 0.97] | 0.79 [0.57, 1.10] | 0.59 [0.11, 3.23] |

*** p<0.001,

** p<0.01,

* p<0.05,

AOR: adjusted odds ratio, CI: Confidence Interval.

†This information was only available for ever-married youth in MDHS and analyzed among ever-married youth in logistic regression.

Philippines (10% in 2017) [23], and Nepal (14.6% in 2016) [24]. The reason for low modern contraceptive utilization might be due to insufficient knowledge of reproductive health, difficulties in accessing quality reproductive health services due to social, financial, or geographical inequalities [13, 25]. Although overall utilization was low, the mCPR among ever-married

youth was acceptable, i.e., 52.2% among all ever-married youth, 44.7% for ever-married male youth, and 54% for ever-married female youth. This finding was almost the same with the overall mCPR of reproductive Myanmar women 15–49 years old (51%) [10] and mCPR of sexually active young women from South Africa (52.2%) [26]. It was also higher than Ethiopia married young women (35.6%) [27]. The mCPR of ever-married youth and female youth reached the target stated in Myanmar's Five-Year Strategic Plan for Young People's Health (2016–2020) to increase CPR among sexually active young people from 38% in 2014 to 52% in 2018 [13].

Interestingly, almost all never-married female youth did not use contraception, and they reported as sexually inactive. Myanmar culture and traditional norms seem to play an important role in preventing premarital sex among never-married youth. This fact might be a possible explanation for very low utilization among this group. Another possible reason was that some unmarried youth might be reluctant to consult with health care providers regarding contraceptive services due to promiscuity issues. Hence, youth-friendly health services would be needed to provide effective reproductive health services primarily to never-married youth [21].

The most commonly used modern contraceptive method was injectable contraception followed by oral contraceptive pills among ever-married youth. In contrast, oral contraceptive pills were mostly used method among never-married youth. This finding was similar to modern contraceptive methods used mainly by Ethiopian youth [28], Nepal youth [29], and study over 21 papers of low-and-middle-income countries [30]. These methods were used mainly by youth due to being easily accessible, available, affordable in almost all private and public clinics, and easy to use. Our study found that utilizing long-term contraceptive methods such as implants and intrauterine devices was very low among youth in Myanmar. Hence, accessibility to long-term contraceptive methods should be promoted free of charge at nearby township health centers for ever-married youth with no desire for more children [31]. All reported contraceptive methods by male youth were female contraceptive methods. Moreover, only two married female youth reported condoms as a modern contraceptive method. This finding pointed out that female takes the primary responsibility for contraception, especially for married youth.

A few youths never heard of contraceptives. Among 11 methods of modern contraception, youth mostly knew 4 to 6 methods. This finding is almost the same as the DHS analysis of men from 18 countries in Asia, Africa, the Caribbean, and Latin America. In this study, most knew 4.5 to 8.8 contraceptive methods, including traditional methods [32]. The male condom was a mostly known method, followed by oral contraceptive pill (OC pill) and injectable contraception among never-married male youth. For ever-married youth, injectable contraception was mostly known, followed by OC pill and male condoms. For all female youth, injectable contraception was mostly known, followed by oral contraceptive pills and female sterilization methods. Being easily accessible and increased health education on how to use those methods are supposed to be the reasons for mostly being known among Myanmar youth [33]. Similarly, the same methods with different orders can be found in the DHS study of Ethiopia youth. Oral contraceptive pills were a mostly known method, followed by injectable contraception and male condoms [28].

We found that the older youth had significantly higher utilization than the younger age group among total male youth and female youth. This finding was consistent with the Ethiopian studies [27, 28], Ghana study [34], and Bangladesh study [35]. However, we did not find this age effect when we analyzed it based on marital status. This fact pointed out that the youth from the 20–24 years age group were more likely to be married, and these married youth utilized more contraception than unmarried youth. Moreover, older youth are more

knowledgeable, employed, and affordable for contraception than younger youth, which might also be a possible reason for higher utilization. Our study did not find a significant association between education and modern contraceptive utilization. The same finding can be seen in the Myanmar study of men [15], even though many other studies had pointed out that youth with higher education had higher utilization of modern contraception [22, 27, 34].

Our study could not provide a significant association between employment status and utilization of modern contraception. This finding was consistent with the Afghanistan study [36]. This finding might be because the reproductive health program had reached out to all youth regardless of the education status and employment status through media or peer education. Youth who had previous exposure to family planning messages from TV, radio, newspaper, internet, billboard, or health care providers did not significantly influence the utilization of modern contraception. This finding was the same with the Myanmar study of men [15] and Afghanistan study [36]. It might be due to getting the family planning information from other sources such as friends, hearsays, or relatives [31, 37]. It is important not to get the wrong information from unreliable sources. Revitalization Youth Information Corner, promoting reproductive health literacy in the community, and a life-skill curriculum at school would increase the reproductive health knowledge of Myanmar youth, ensuring safer sex among youth in Myanmar [13, 21].

Sexual activity significantly influenced the utilization of modern contraception. The utilization among sexually active youth was significantly higher than that of not sexually active youth, and this finding was consistent with a study in Burkina Faso and Mali [22]. Ever-married youth who did not have a desire for more children significantly utilized more modern contraception than those who wanted more children, and the same finding can be seen in Nepal study [29], Ethiopia study [27], Senegal study [38], and Myanmar study [16].

We found the dose-response relationship between knowledge of known methods and utilization of modern contraception. As increased in the number of known modern contraceptive methods, modern contraceptive utilization also increased. The same finding can be seen in the Senegal study [38], Ethiopia study [39], and Nigeria study [40]. It might be due to youth with high knowledge of contraception having a better choice of modern contraception and more likely to have a positive attitude towards using modern contraception [37].

Youth from the rural areas had significantly lower utilization of modern contraception than youth from urban areas among ever-married youth, female youth, and ever-married female youth. This finding was consistent with the Bangladesh study [35] and West Africa study [22]. However, the residence was not a significant predictor for male youth and never-married youth. The same finding can be seen in the Myanmar men study [15] and the study conducted among men from Asia, Africa, the Caribbean, and Latin America [32]. The difference in accessibility and affordability between urban and rural areas might explain the low utilization of modern contraception among youth from rural areas.

Youth from the hilly, coastal, and central plain zones had significantly lower utilization of modern contraception than youth from the delta zone. This finding was consistent with the Myanmar study [15]. Delta zone includes Yangon, Ayeyarwady, and Bago regions which are the most developed regions in Myanmar. Hence, better accessibility, availability, and affordability of contraception in the delta zone than in other zones might be a possible explanation of regional variation in contraceptive utilization. Current implementing reproductive health programs and services should be equally accessible without geographic variation.

Household wealth status influenced the utilization of contraception only among female youth. Youth from rich households utilized modern contraception less than those from poor households, and it was not consistent with the Ethiopian study [27], Afghanistan study [36], and West Africa study [22]. However, this effect was diminished as we analyzed separately for

married and unmarried youth. In Myanmar, both government and non-governmental organizations' reproductive health clinics provide reproductive health services free of charge or low costs. Hence, wealth status could not be a barrier to modern contraceptive utilization among Myanmar youth.

## Strengths and limitations of the study

We used nationally representative data from MDHS (2015–2016), and all analyses were weighted to get national estimates. Therefore, the generalizability of the findings from this study is high. The findings from this study pointed out the country's contraceptive prevalence rate among youth and factors influencing those conditions, which might help promote youth's family planning health services by pointing out the area to emphasize.

Although all interviews had been conducted by enumerator-respondent match in terms of gender at a place where privacy was ensured, some cultural barriers might be possible for underestimating contraceptive utilization among never-married youth. The causality cannot be applied because of being data from a cross-sectional survey. Some predictors supported by literature could not be included in this study due to data limitations. Hence, we could not provide evidence of bad outcomes due to not using modern contraception and the success stories of using modern contraception. We could not also assess whether youth who did not use modern contraception leave the school earlier than they would have, get unplanned pregnancy and sexually transmitted infection due to unsafe sex. Although we found female takes the primary responsibility for contraception, our study could not provide the evidence of male participation in the choice of modern contraception. Moreover, we used the data from MDHS (2015–2016); the modern contraceptive prevalence rates and the number of modern contraceptive methods known by youth might be changed during these years. However, the determinants of modern contraceptive utilization might be the same and applicable for program implementation.

## Conclusion

The utilization of modern contraception of Myanmar youth was low. The knowledge on modern contraceptive methods favored the utilization. Some youth did not know any modern contraception methods, which was an alarm sign to program implementers to promote reproductive health education effectively among the youth population. The reproductive health program should emphasize the adolescents and never-married youth population, especially in areas with low utilization, such as from the hilly and rural areas, to have equitable access to quality reproductive health services and health literacy. Further studies using the mixed method approach should be conducted to explore the barriers and challenges of contraceptive utilization and male involvement in the choice of contraception among youth. Moreover, revitalization of Youth Information Corner (YIC) and youth-friendly reproductive health services are needed to increase reproductive health knowledge and utilization to prevent unsafe sex, unintended pregnancies, and abortions which might help in reducing maternal and child mortality.

## Acknowledgments

We want to thank the DHS Program (ICF) for permitting MDHS data access for this study. We also want to express our sincere gratitude to the Institutional Review Board, University of Public Health, Yangon, for giving ethical clearance to conduct this study.

## Author Contributions

**Conceptualization:** Ciin Ngaih Lun, Thida Aung, Kyaw Swa Mya.

**Data curation:** Ciin Ngaih Lun, Thida Aung, Kyaw Swa Mya.

**Formal analysis:** Ciin Ngaih Lun, Kyaw Swa Mya.

**Methodology:** Ciin Ngaih Lun, Thida Aung, Kyaw Swa Mya.

**Software:** Ciin Ngaih Lun, Kyaw Swa Mya.

**Supervision:** Thida Aung, Kyaw Swa Mya.

**Writing – original draft:** Ciin Ngaih Lun.

**Writing – review & editing:** Thida Aung, Kyaw Swa Mya.

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
