## [Decision Letter · Decision Letter 0]

8 Jun 2021

PONE-D-21-13352

Utilization of modern contraceptive methods and its determinants among youth in Myanmar: Further analysis of Myanmar Demographic and Health Survey (2015-2016)

PLOS ONE

Dear Dr. Mya,

Thank you for submitting your manuscript to PLOS ONE. After careful consideration, we feel that it has merit but does not fully meet PLOS ONE’s publication criteria as it currently stands. Therefore, we invite you to submit a revised version of the manuscript that addresses the points raised during the review process.

Given the situation in Myanmar, I want to give you as much flexibility as possible without limitations or deadlines, but I want to suggest the most important changes that will help improve this paper so that we can publish it. See below for how to request more time. 

Required changes:

1. Marital status is clearly such a large determinant of contraceptive use that it makes it hard to see any other effects, and the odds ratios are enormous and the confidence intervals are wide.  Please stratify the analysis into married and unmarried to avoid these enormous odds ratios and wide intervals. That is, married females, unmarried females, married males, unmarried males.

2. Studies of contraceptive use generally use a health behavior model to guide studies and later to guide interventions. Examples of health behavior models used to study contraception include social cognitive theory, health belief model, theory of planned behavior, and protection motivation theory. For example, this paper (available on researchgate without a paywall) uses the health behavior model.

Mon MM, Liabsuetrakul T. Predictors of contraceptive use among married youths and their husbands in a rural area of Myanmar. Asia Pac J Public Health. 2012 Jan;24(1):151-60. doi: 10.1177/1010539510381918. Epub 2010 Sep 9. PMID: 20829275. Choose a health behavior model and use it to guide the study and choose important variables. Some important variables could be self-efficacy to use a method, perceived risk of pregnancy, not wanting pregnancy now, etc., if they are in the DHS. 3. Knowledge of contraception methods (or if it were important exposure to family planning messages) may be an intermediate variable because it's impossible to use a method without knowledge that the method exists. The choice to divide into no, 1-5, and 5+ seems unrelated to practice, however. In practice, someone just needs to know 1 modern method in order to use it, and knowing 5 methods isn't necessarily better than knowing 4 methods or 3 methods, as long as they have access and can use one method. Also, some of these methods in the list aren't useful for childless people with future fertility intentions: female and male sterilization, lactational amenorrhea. Also, emergency contraception isn't a first line contraception, so maybe a different category. Consider reconceptualizing the variable to address the methods that would be most important for people of this age and life stage. If you think this variable is important, consider evaluating whether knowledge of contraceptive methods is a mediating variable using a mediation method. The medeff command in Stata by Dustin Tingley and Raymond Hicks will do one type of mediation analysis from  Imai, Keele, and Tingley that is easy to interpret.  4. The study should describe practical implications of this study. You may need to do additional analyses to answer this question.  These are some suggestions:a. Based on analysis of these data, what are the bad outcomes of not using modern contraception? That is, among the married and unmarried young women in this dataset who are not using modern contraception, can you find any outcomes that might be the result of not using modern contraception?  Do you see evidence that they're leaving education earlier than they would have otherwise, having unplanned pregnancies, have bad marriages that they can't leave due to children, have STIs? b. Within these data, who are the success stories? Who are the people who would be expected not to use contraception but nonetheless are using modern contraception?  Can we learn from them? Can these people point to what interventions could be done to improve contraception access for everyone? c. You mention Youth Information Corner in the conclusion without explaining what this is and why it's important.  It should be mentioned earlier and described and explain how to improve it to help both married and unmarried young people. Based on this research, what should be done differently in order to improve access to contraception, knowledge of contraception, and acceptance of contraception?  What populations and which ages could be targeted with which interventions? If you were going to write a grant to propose a program or intervention, what would you propose? If these interventions were done, what positive impacts would there be? d. It looks like exposure to family planning messages aren't associated with greater use --- does that mean that any message-based interventions aren't likely to be effective because they rely on exposure to family planning messages? e.  Based on this study, what should be looked for in Myanmar's 2021 DHS that seems to be currently in the field? If you were going to add questions to the DHS in the future, what would you want to know? What study will you do using this 2021 DHS? Are there any policies that have happened between these two studies that might predict a change in the associations you have observed?f. Males are less aware of contraception options, and they are less likely to report use of certain methods such as injectable contraception, which may mean that they just didn't know that their female partner was getting them.  Are the males taking enough responsibility here? How should interventions target males to involve them in contraception choices? What barriers need to be overcome to include males more and to improve their motivation to learn about contraception? Do you see evidence that any males are possibly preventing female partners from accessing contraception?  Can you find any attitudes measured on DHS that explain which males are more involved in contraception choices? 5.  Stepwise regression is not considered a rigorous method for finding important variables because it ends up just finding the p-values that are significant, which may differ across datasets. That is, if you repeated this analysis in the 2021 Myanmar DHS, you might find other variables are important. Using a cross-validation approach of choosing the variables based on analysis of another dataset (e.g., is there an earlier DHS?) and then testing the variables in this dataset is more rigorous. An alternative method is to identify the most important variables based on theory, add them in order of importance, and keep them in even if not significant if they aren't hurting the model. To the extent feasible, identify a method for variable choice other than stepwise regression. Andrew Gelman writes about stepwise regression in his blog here and the comments have a lot of resources: https://statmodeling.stat.columbia.edu/2014/06/02/hate-stepwise-regression/ Suggested revisions:1.  If possible, Figure 1 would be nicer to show histogram of the number of methods, not the 3 broad categories. Even better, consider a dot plot (Cleveland's dot plot, seems to be graph dot in stata) to show the gap of male married vs female married knowledge of each method, and another dot plot showing the gap between male unmarried vs. female unmarried.  The specific method information is really much more interesting than the number. It's also possibly more actionable.

We look forward to receiving your revised manuscript.

Kind regards,

Janet E Rosenbaum, Ph.D.

Academic Editor

PLOS ONE

Journal Requirements:

2.Please provide additional details regarding participant consent. In the ethics statement in the Methods and online submission information, please ensure that you have specified (1) whether consent was informed and (2) what type you obtained (for instance, written or verbal, and if verbal, how it was documented and witnessed). If your study included minors, state whether you obtained consent from parents or guardians. If the need for consent was waived by the ethics committee, please include this information.

Reviewers' comments:

Reviewer's Responses to Questions

**Comments to the Author**

1. Is the manuscript technically sound, and do the data support the conclusions?

Reviewer #1: Yes

Reviewer #2: Partly

2. Has the statistical analysis been performed appropriately and rigorously? 

Reviewer #1: Yes

Reviewer #2: Yes

3. Have the authors made all data underlying the findings in their manuscript fully available?

Reviewer #1: Yes

Reviewer #2: Yes

4. Is the manuscript presented in an intelligible fashion and written in standard English?

Reviewer #1: Yes

Reviewer #2: Yes

5. Review Comments to the Author

Reviewer #1: Title

This paper appears one of important papers about sexual and reproductive health among Myanmar youth. However, why use the word “further analysis”? Was there any similar paper about contraceptive methods/practices?

Introduction

This session highlighted the linkage between utilization of contraceptive methods and low maternal and neonatal deaths. In Myanmar, is there any study about the utilization of contraceptive methods among youth or general population? If no, it should specify in introduction session because it is one of knowledge gaps for this study.

Research aim

The research aim well reflects the purpose of this study. But, one thing, because it mentioned to meet the mandates of policies and program of Myanmar, what are the mandates, current situation, and gaps of policy implementation? It should mention in introduction session.

Methodology

The use of cross-sectional design is appropriate for this study. The sampling method is clear. However, is there any inclusion criteria or exclusion criteria for sample selection? Giving detail explanation will help the readers in understanding the research results.

Line 114-115: “Women's employment variable was recoded as "yes" if he or she currently had an earning job”. It makes a little confused why the man with an earning job is recorded as “yes” for women’s employment. Does it mean the husband/man has a job to be recorded as women’s employment, not wife/woman?

Statistical Method

The use of STATA software is appropriate for this study.

What motivates to use “age variables” into categorized group, rather than continuous variables? Is there any change if the age variable is applied as a continuous one in the regression analysis?

Results

The finding represents the statistical analysis performed by the authors.

Report mean age and standard deviation for each group of age.

Just a curiosity, why did the analysis not perform for never-married youth?

Discussion

This study well discussed about the findings by using comparative global studies and sound theoretical background.

Additionally, it should highlight some interesting facts such as male participants did not use male condoms (which is the main contraceptive method for men, and this is one important issue for program implementers.) and about 7 percent of study participants did not know any modern contraceptive method (it means about 93 percent of participants knew at least one contraceptive method. Is it the achievement of health education program?)

For limitation of study, is there any bias during interviews? For e.g., the informants selectively response the questions such as not telling the use of contraception due to cultural barriers especially for unmarried youth.

Conclusion

The study could be improved by clarifying the gaps between current program implementation and the findings of the study. The authors’ recommendations are appropriate, but it would be more effective if the study suggest the supportive role of society to create a sound environment in utilization of contraceptive methods. For e.g. The family planning program should expand its health education sessions to not only married couples but also all reproductive ages in the country. It is just optional.

Grammatical errors

Please review the manuscript for grammatical and spelling errors.

Recommendation

I would like to recommend acceptance of this paper after the authors have reviewed and addressed the above suggestions.

Reviewer #2: The study's purpose and analysis are terrific. The introduction is well-highlighted that it is crucial to determine social determinants of high teen birth rates, especially in developing countries.

Table 4 should state the statistical method used for the analysis to find more helpful for the audience.

Some interpretations should use " how many times higher or lower" instead of jargon (aOR) in results secession. Then, the author should state controlling other covariates while interpreting the multivariate logistic regression analysis.

Knowledge of the number of contraception methods " Known modern methods" showed a trend of dose-response relationship supporting a stronger causal relationship between knowledge of the number of contraception methods and utilization of contraception.

In discussion, the first few sentences are reiterated the above findings. Then, emphasizing the dose-response relationship will add a more powerful point to convince the conclusions of " The knowledge on modern contraceptive methods favored the utilization." The present study's findings agreed to similar findings that the authors referenced. (Reference no 18).

Strength and limitation: some of the final models in the male and ever-married population did not reach significant findings, probably due to a small sample size ( male = 1,423 and ever married = 1,424).

6. PLOS authors have the option to publish the peer review history of their article (what does this mean?). If published, this will include your full peer review and any attached files.

Reviewer #1: No

Reviewer #2: **Yes: **Phyu T. Mar

---

## [Author Response · Author response to Decision Letter 0]

22 Jul 2021

Responses to Academic Editor

Dear Dr. Janet E Rosenbaum

We appreciate your insightful comments and for giving valuable suggestions that improve our paper a lot. 

1. Marital status is clearly such a large determinant of contraceptive use that it makes it hard to see any other effects, and the odds ratios are enormous and the confidence intervals are wide. Please stratify the analysis into married and unmarried to avoid these enormous odds ratios and wide intervals. That is, married females, unmarried females, married males, unmarried males.

Author's response – We followed your suggestion, and separate analyses were done for married females, unmarried females, married males, unmarried males. Please see in the results section (Lines 174-301).

2. Studies of contraceptive use generally use a health behavior model to guide studies and later to guide interventions. Examples of health behavior models used to study contraception include social cognitive theory, health belief model, theory of planned behavior, and protection motivation theory. For example, this paper (available on researchgate without a paywall) uses the health behavior model.

Mon MM, Liabsuetrakul T. Predictors of contraceptive use among married youths and their husbands in a rural area of Myanmar. Asia Pac J Public Health. 2012 Jan;24(1):151-60. doi: 10.1177/1010539510381918. Epub 2010 Sep 9. PMID: 20829275.

Choose a health behavior model and use it to guide the study and choose important variables. Some important variables could be self-efficacy to use a method, perceived risk of pregnancy, not wanting pregnancy now, etc., if they are in the DHS.

Author's response – Thank you for providing this good article. We cited this article in our manuscript with reference number [20]. 

Since we are conducting secondary data analysis using DHS data, we don't have enough variables to measure the health belief model's constructs – perceived susceptibility, severity, barriers, benefit, efficacy, and cue to action.

However, we added two more variables that might influence utilization – sexual activity and desire for more children. Please see Lines 139-142.

3. Knowledge of contraception methods (or if it were important exposure to family planning messages) may be an intermediate variable because it's impossible to use a method without knowledge that the method exists. The choice to divide into no, 1-5, and 5+ seems unrelated to practice, however. In practice, someone just needs to know 1 modern method in order to use it, and knowing 5 methods isn't necessarily better than knowing 4 methods or 3 methods, as long as they have access and can use one method. Also, some of these methods in the list aren't useful for childless people with future fertility intentions: female and male sterilization, lactational amenorrhea. Also, emergency contraception isn't a first line contraception, so maybe a different category. Consider reconceptualizing the variable to address the methods that would be most important for people of this age and life stage.

If you think this variable is important, consider evaluating whether knowledge of contraceptive methods is a mediating variable using a mediation method. The medeff command in Stata by Dustin Tingley and Raymond Hicks will do one type of mediation analysis from Imai, Keele, and Tingley that is easy to interpret. 

Author's response – According to your suggestion, we did not divide the knowledge into no, 1-5, and 5+. Instead, we used it as a linear score range 0-11. Please see Line 127-128, Table 5, and Table 6.

We follow the Myanmar Demographic and Health Survey annual report [10], in which all these 11 methods are considered modern contraceptive methods. https://mohs.gov.mm/cat/MDHS%20(2015-16) We assessed their knowledge for these contraceptive methods; however, for utilization, we interpret the availability and the most commonly used methods among youth.

We used the knowledge variable as one of the independent variables rather than a mediating variable.

4. The study should describe practical implications of this study. You may need to do additional analyses to answer this question. These are some suggestions:

a. Based on analysis of these data, what are the bad outcomes of not using modern contraception? That is, among the married and unmarried young women in this dataset who are not using modern contraception, can you find any outcomes that might be the result of not using modern contraception? Do you see evidence that they're leaving education earlier than they would have otherwise, having unplanned pregnancies, have bad marriages that they can't leave due to children, have STIs? 

Author's response – Since we conduct secondary data analysis using MDHS data, some variables that need to answer your questions are not available in the data. Hence, we added these limitations to the discussion. Please see lines 411-416.

b. Within these data, who are the success stories? Who are the people who would be expected not to use contraception but nonetheless are using modern contraception? Can we learn from them? Can these people point to what interventions could be done to improve contraception access for everyone? 

Author's response – Since we conduct secondary data analysis using MDHS data, some variables that need to answer your questions are not available in the data. Hence, we added these limitations to the discussion. Please see lines 411-416.

c. You mention Youth Information Corner in the conclusion without explaining what this is and why it's important. It should be mentioned earlier and described and explain how to improve it to help both married and unmarried young people. Based on this research, what should be done differently in order to improve access to contraception, knowledge of contraception, and acceptance of contraception? What populations and which ages could be targeted with which interventions? If you were going to write a grant to propose a program or intervention, what would you propose? If these interventions were done, what positive impacts would there be? 

Author's response – We added a paragraph about Youth Information Corner. Please see lines 88-95. 

 For other questions, we gave these recommendations according to our findings.

• The reproductive health program should emphasize the adolescents and never-married youth population, especially in areas with low utilization, such as from the hilly and rural areas, to have equitable access to quality reproductive health services and health literacy. 

• Moreover, revitalization of Youth Information Corner (YIC) and youth-friendly reproductive health services are needed to increase reproductive health knowledge and utilization to prevent unsafe sex, unintended pregnancies, and abortions which might help in reducing maternal and child mortality."

d. It looks like exposure to family planning messages aren't associated with greater use --- does that mean that any message-based interventions aren't likely to be effective because they rely on exposure to family planning messages? 

Author's response – We explain this finding in lines 360-367.

e. Based on this study, what should be looked for in Myanmar's 2021 DHS that seems to be currently in the field? If you were going to add questions to the DHS in the future, what would you want to know? What study will you do using this 2021 DHS? Are there any policies that have happened between these two studies that might predict a change in the associations you have observed?

Author's response – Although we planned to conduct the second DHS study in 2020, the global COVID-19 pandemic interfered with the survey and we cannot start conducting this survey the second time. We would like to add some questions you pointed out, such as bad and success stories of utilization, males' role in contraceptive utilization. 

Some of these questions could not get the detailed answers using the quantitative survey alone; hence, we gave research implication that "Further studies using the mixed method approach should be conducted to explore the barriers and challenges of contraceptive utilization and male involvement in choosing contraception among youth." Please see Lines 428-430.

f. Males are less aware of contraception options, and they are less likely to report use of certain methods such as injectable contraception, which may mean that they just didn't know that their female partner was getting them. Are the males taking enough responsibility here? How should interventions target males to involve them in contraception choices? What barriers need to be overcome to include males more and to improve their motivation to learn about contraception? Do you see evidence that any males are possibly preventing female partners from accessing contraception? Can you find any attitudes measured on DHS that explain which males are more involved in contraception choices?

Author's response – We found very low utilization of condoms among both never-married and ever-married youth. This finding pointed out that female youth took responsibility for contraception. We discuss this finding in lines 332-335.

5. Stepwise regression is not considered a rigorous method for finding important variables because it ends up just finding the p-values that are significant, which may differ across datasets. That is, if you repeated this analysis in the 2021 Myanmar DHS, you might find other variables are important. Using a cross-validation approach of choosing the variables based on analysis of another dataset (e.g., is there an earlier DHS?) and then testing the variables in this dataset is more rigorous. An alternative method is to identify the most important variables based on theory, add them in order of importance, and keep them in even if not significant if they aren't hurting the model. To the extent feasible, identify a method for variable choice other than stepwise regression. Andrew Gelman writes about stepwise regression in his blog here and the comments have a lot of resources: 

https://statmodeling.stat.columbia.edu/2014/06/02/hate-stepwise-regression/

Author's response – Thank you for pointing out the weakness of the statistically driven model. Hence, we chose the variables based on the literature search in which these variables were associated with the utilization. Then, we reanalyzed the theoretical driven model by using the enter method not removing any variable (including all variables found in the literature associated with outcome and available in data). Please see Table 5 and Table 6. 

Suggested revisions:

1. If possible, Figure 1 would be nicer to show histogram of the number of methods, not the 3 broad categories. Even better, consider a dot plot (Cleveland's dot plot, seems to be graph dot in stata) to show the gap of male married vs female married knowledge of each method, and another dot plot showing the gap between male unmarried vs. female unmarried. The specific method information is really much more interesting than the number. It's also possibly more actionable.

Author's response – We revised Figure 1 and described it by mirror bar chart using Excel. Please see Fig 1.tiff file.

Author's response – We added some sentence for the consent taking and data anonymity issue. Please see lines 169-170.

Responses to Reviewer 1

Dear Reviewer

Thank you very much for you appreciations and thorough comments. We followed your comments and revised the manuscript.

1. This paper appears one of important papers about sexual and reproductive health among Myanmar youth. However, why use the word "further analysis"? Was there any similar paper about contraceptive methods/practices?

Author's response – We remove the word "Further" in the title. Please see the title.

2. Introduction

This session highlighted the linkage between utilization of contraceptive methods and low maternal and neonatal deaths. In Myanmar, is there any study about the utilization of contraceptive methods among youth or general population? If no, it should specify in introduction session because it is one of knowledge gaps for this study.

Author's response – We add a paragraph for previous studies and knowledge gaps for this study. Please see lines 96-102.

3. Research aim

The research aim well reflects the purpose of this study. But, one thing, because it mentioned to meet the mandates of policies and program of Myanmar, what are the mandates, current situation, and gaps of policy implementation? It should mention in introduction session.

Author's response – We add a paragraph regarding the Five-Year Strategic Plan for Young People's Health (2016-2020). Please see Lines 79-87.

4. Methodology

The use of cross-sectional design is appropriate for this study. The sampling method is clear. However, is there any inclusion criteria or exclusion criteria for sample selection? Giving detail explanation will help the readers in understanding the research results.

Author's response – I added selection criteria of the DHS survey in "Materials and Methods" section. Please see lines 110-114.

5. Line 114-115: "Women's employment variable was recoded as "yes" if he or she currently had an earning job". It makes a little confused why the man with an earning job is recorded as "yes" for women's employment. Does it mean the husband/man has a job to be recorded as women's employment, not wife/woman?

Author's response – We revised the sentence like that "The employment variable was recoded as "yes" if the respondent had an earning job within 12 months before the survey or "no" otherwise." Please see the line 135-136.

6. Statistical Method

The use of STATA software is appropriate for this study.

What motivates to use "age variables" into categorized group, rather than continuous variables? Is there any change if the age variable is applied as a continuous one in the regression analysis?

Author's response –Although we used age as a continuous variable, the findings were not changed in direction and its significant effect. All DHS annual reports categorized age into 5 years intervals. We follow this rule to be easier to compare with other countries' studies and reports. Hence, we categorized age into 15-19 and 20-24. Please check the reference [10]. 

7. Results

The finding represents the statistical analysis performed by the authors.

Report mean age and standard deviation for each group of age.

Just a curiosity, why did the analysis not perform for never-married youth?

Author's response – We reported mean age and standard deviation in Table 1 (Line 188). Now we added the findings of never-married youth in Table 6 (Line 300).

8. Discussion

This study well discussed about the findings by using comparative global studies and sound theoretical background.

Additionally, it should highlight some interesting facts such as male participants did not use male condoms (which is the main contraceptive method for men, and this is one important issue for program implementers.) and about 7 percent of study participants did not know any modern contraceptive method (it means about 93 percent of participants knew at least one contraceptive method. Is it the achievement of health education program?)

For limitation of study, is there any bias during interviews? For e.g., the informants selectively response the questions such as not telling the use of contraception due to cultural barriers especially for unmarried youth.

Author's response – We added some sentences regarding male condoms utilization and responsibility regarding contraception among youth. Please see lines 332-335. 

We mention the possibility of underestimation of utilization due to cultural barriers among unmarried youth in limitations. Please see lines 408-410.

9. Conclusion

The study could be improved by clarifying the gaps between current program implementation and the findings of the study. The authors' recommendations are appropriate, but it would be more effective if the study suggest the supportive role of society to create a sound environment in utilization of contraceptive methods. For e.g. The family planning program should expand its health education sessions to not only married couples but also all reproductive ages in the country. It is just optional.

Author's response – We would like to give some suggestions in conclusion as follow:

 "The reproductive health program should emphasize the adolescents and never-married youth population, especially in areas with low utilization, such as from the hilly and rural areas, to have equitable access to quality reproductive health services and health literacy. Moreover, revitalization of Youth Information Corner (YIC) and youth-friendly reproductive health services are needed to increase reproductive health knowledge and utilization to prevent unsafe sex, unintended pregnancies, and abortions which might help in reducing maternal and child mortality."

Please see lines 426-433. 

10. Grammatical errors

Please review the manuscript for grammatical and spelling errors.

Author's response –The manuscript was proofread to check grammatical and spelling errors.

11. Recommendation

I would like to recommend acceptance of this paper after the authors have reviewed and addressed the above suggestions.

Author's response – Thank you very much for your kind considerations.

Responses to Reviewer 2

Dear Reviewer

Thank you very much for you appreciations and thorough comments. We followed your comments and revised the manuscript.

1. The study's purpose and analysis are terrific. The introduction is well-highlighted that it is crucial to determine social determinants of high teen birth rates, especially in developing countries.

Table 4 should state the statistical method used for the analysis to find more helpful for the audience.

Some interpretations should use " how many times higher or lower" instead of jargon (aOR) in results secession. Then, the author should state controlling other covariates while interpreting the multivariate logistic regression analysis.

Author's response – We added the statistical method "Chi-square test for independence" in the Methods section and footnote of Table 4. Please see lines 152-153 and lines 245-246. 

We remove the jargon in the interpretation. Please see lines 247-266 and line 272-299.

We add the phrase "adjusting the covariates" to interpret the multivariable binary logistic regression analysis. Please see lines 249 and line 273.

2. Knowledge of the number of contraception methods "Known modern methods" showed a trend of dose-response relationship supporting a stronger causal relationship between knowledge of the number of contraception methods and utilization of contraception.

Author's response – We added some sentences about the dose-response relationship of knowledge of modern contraceptive methods in the discussion. Please see lines 373-378.

3. In discussion, the first few sentences are reiterated the above findings. Then, emphasizing the dose-response relationship will add a more powerful point to convince the conclusions of "The knowledge on modern contraceptive methods favored the utilization." The present study's findings agreed to similar findings that the authors referenced. (Reference no 18).

Author's response – We remove the reiterated findings in the discussion.

4. Strength and limitation: some of the final models in the male and ever-married population did not reach significant findings, probably due to a small sample size ( male = 1,423 and ever married = 1,424).

Author's response – Thanks for giving a possible explanation for our findings.

---

## [Decision Letter · Decision Letter 1]

3 Sep 2021

PONE-D-21-13352R1

Utilization of modern contraceptive methods and its determinants among youth in Myanmar: Analysis of Myanmar Demographic and Health Survey (2015-2016)

PLOS ONE

Dear Dr. Mya,

Thank you for submitting your manuscript to PLOS ONE. After careful consideration, we feel that it has merit but does not fully meet PLOS ONE’s publication criteria as it currently stands. Therefore, we invite you to submit a revised version of the manuscript that addresses the points raised during the review process.

We look forward to receiving your revised manuscript.

Kind regards,

Janet E Rosenbaum, Ph.D.

Academic Editor

PLOS ONE

Journal Requirements:

Additional Editor Comments (if provided):

Reviewers' comments:

Reviewer's Responses to Questions

**Comments to the Author**

1. If the authors have adequately addressed your comments raised in a previous round of review and you feel that this manuscript is now acceptable for publication, you may indicate that here to bypass the “Comments to the Author” section, enter your conflict of interest statement in the “Confidential to Editor” section, and submit your "Accept" recommendation.

Reviewer #1: All comments have been addressed

Reviewer #2: All comments have been addressed

2. Is the manuscript technically sound, and do the data support the conclusions?

Reviewer #1: Yes

Reviewer #2: Yes

3. Has the statistical analysis been performed appropriately and rigorously? 

Reviewer #1: Yes

Reviewer #2: Yes

4. Have the authors made all data underlying the findings in their manuscript fully available?

Reviewer #1: Yes

Reviewer #2: Yes

5. Is the manuscript presented in an intelligible fashion and written in standard English?

Reviewer #1: Yes

Reviewer #2: Yes

6. Review Comments to the Author

Reviewer #1: The authors well responded to all comments of previous review. Thank you for well addressing them. However, after this current review, some minor comments came out to be addressed by the authors. Please find the following:

1. Line 127-128 Why was “the number of modern contraceptive methods known” not categorized? I think it was because reviewers suggested revising the category of this variable in the first review. In Table 5 and Table 6 of the revised manuscript, the value of this variable did not describe any reference category. If it intended to use continuous variable, the result of logistic regression on this continuous variable should be completely described in Table 5 and Table 6. On the other hand, if it used binary variable, the variable should be categorized into at least 2 categories. For example, “0” for nothing known, and “1-11” for knowing one and more methods or “no”/ “yes” category.

2. In Line 338, it described “most knew 4.5 to 8.8 contraceptive methods”. Numerically, it is true to describe the real value. But logically, the method should be counted in compete number. 4.5 or 8.8 Methods looks a little unrealistic. One method cannot be divided into 0.5 or 0.8 method. This is just optional.

3. Line 373 described dose-response relationship between knowledge and utilization. However, the manuscript did not describe the statistical values of this relationship in the result session. May be I misunderstood Table 6. My suggestion is to use a figure or illustration to describe “this dose-response relationship” for more understanding by readers.

4. Line 392-393 argued that youth from rich households utilized more contraceptives than ones from poor household. It looks contradictory to the finding. In Line 287, it stated that “Female youth from rich households were 32% less likely to use modern contraception than poor households”. Please review it and kindly keep consistency of manuscript.

That’s all from my side and I would like to recommend acceptance of this manuscript after the authors have addressed the minor suggestions.

Reviewer #2: (No Response)

7. PLOS authors have the option to publish the peer review history of their article (what does this mean?). If published, this will include your full peer review and any attached files.

Reviewer #1: No

Reviewer #2: No

---

## [Author Response · Author response to Decision Letter 1]

6 Sep 2021

Responses to Reviewer 1

Dear Reviewer

Thank you very much for pointing out some mistakes of us. I revised according to your comments.

1. Line 127-128 Why was “the number of modern contraceptive methods known” not categorized? I think it was because reviewers suggested revising the category of this variable in the first review. In Table 5 and Table 6 of the revised manuscript, the value of this variable did not describe any reference category. If it intended to use continuous variable, the result of logistic regression on this continuous variable should be completely described in Table 5 and Table 6. On the other hand, if it used binary variable, the variable should be categorized into at least 2 categories. For example, “0” for nothing known, and “1-11” for knowing one and more methods or “no”/ “yes” category.

Author's response – We received the comments from the Academic editor not to categorize the number of modern contraceptive methods known; hence, we revised accordingly and treated it as a continuous variable. To clear the reader, we added a sentence at Line 128-129 "we treated this variable as a continuous variable."

Since we treated this variable as a continuous independent variable, we don't need a reference category for binary logistic regression analysis for the dependent variable (contraceptive utilization) in Table 5 and Table 6. We interpreted the results like how much odds of contraceptive utilization were increased or decreased as one more method knew.

2. In Line 338, it described “most knew 4.5 to 8.8 contraceptive methods”. Numerically, it is true to describe the real value. But logically, the method should be counted in compete number. 4.5 or 8.8 Methods looks a little unrealistic. One method cannot be divided into 0.5 or 0.8 method. This is just optional.

Author's response – We agree with your comments since the number of known contraceptive methods was a discrete numerical variable. However, this finding was not our study's finding. We used the finding of reference number [32] to discuss knowledge of modern contraceptive methods. 

3. Line 373 described dose-response relationship between knowledge and utilization. However, the manuscript did not describe the statistical values of this relationship in the result session. May be I misunderstood Table 6. My suggestion is to use a figure or illustration to describe “this dose-response relationship” for more understanding by readers.

Author's response – We performed a chi-square test for trend to describe this dose-response relationship and the results were described using Fig 3. Please see Fig 3 and the description of this figure in Lines 227-230.

4. Line 392-393 argued that youth from rich households utilized more contraceptives than ones from poor household. It looks contradictory to the finding. In Line 287, it stated that “Female youth from rich households were 32% less likely to use modern contraception than poor households”. Please review it and kindly keep consistency of manuscript.

Author's response – Thank you so much for pointing out this gross mistake. We revised this finding appropriately. Please see Line 399.

Other minor corrections by Author

1. In line 184 of the unmarked version, we replaced the words "three fourth" with "three-fourths".

2. In lines 260-261 of the unmarked version, we replaced the sentence "Youth from the hilly zone were 70% fewer odds to use modern contraception than those from the delta zone" with "Youth from the hilly zone had 70% fewer odds of using modern contraception than those from the delta zone".

---

## [Decision Letter · Decision Letter 2]

20 Sep 2021

Utilization of modern contraceptive methods and its determinants among youth in Myanmar: Analysis of Myanmar Demographic and Health Survey (2015-2016)

PONE-D-21-13352R2

Dear Dr. Mya,

We’re pleased to inform you that your manuscript has been judged scientifically suitable for publication and will be formally accepted for publication once it meets all outstanding technical requirements.

Kind regards,

Janet E Rosenbaum, Ph.D.

Academic Editor

PLOS ONE

Additional Editor Comments (optional):

Reviewers' comments:

Reviewer's Responses to Questions

**Comments to the Author**

1. If the authors have adequately addressed your comments raised in a previous round of review and you feel that this manuscript is now acceptable for publication, you may indicate that here to bypass the “Comments to the Author” section, enter your conflict of interest statement in the “Confidential to Editor” section, and submit your "Accept" recommendation.

Reviewer #1: All comments have been addressed

2. Is the manuscript technically sound, and do the data support the conclusions?

Reviewer #1: Yes

3. Has the statistical analysis been performed appropriately and rigorously? 

Reviewer #1: Yes

4. Have the authors made all data underlying the findings in their manuscript fully available?

Reviewer #1: Yes

5. Is the manuscript presented in an intelligible fashion and written in standard English?

Reviewer #1: Yes

6. Review Comments to the Author

Reviewer #1: Dear Authors,

Thanks for addressing all the comments and I really appreciate it.

If you don't mind, I would like to provide this minor suggestion.

Line 401-403 Please review those lines and kindly adjust the meaning of sentence to be consistent with the revision (wealthy youth utilize less modern contraceptives than poor youth). Or, alternatively, it will not be a problem if those lines are deleted.

This is just optional and hope it will be helpful to you.

Thank you.

7. PLOS authors have the option to publish the peer review history of their article (what does this mean?). If published, this will include your full peer review and any attached files.

Reviewer #1: No

---

## [Editor Report · Acceptance letter]

24 Sep 2021

PONE-D-21-13352R2 

Utilization of modern contraceptive methods and its determinants among youth in Myanmar: Analysis of Myanmar Demographic and Health Survey (2015-2016) 

Dear Dr. Mya:

I'm pleased to inform you that your manuscript has been deemed suitable for publication in PLOS ONE. Congratulations! Your manuscript is now with our production department. 

Kind regards, 

on behalf of

Dr. Janet E Rosenbaum 

Academic Editor

PLOS ONE